# Q-SNNs: Quantized Spiking Neural Networks

## ABSTRACT

Brain-inspired Spiking Neural Networks (SNNs) leverage sparse spikes to represent information and process them in an asynchronous event-driven manner, offering an energy-efficient paradigm for the next generation of machine intelligence. However, the current focus within the SNN community prioritizes accuracy optimization through the development of large-scale models, limiting their viability in resource-constrained and low-power edge devices. To address this challenge, we introduce a lightweight and hardware-friendly Quantized SNN (Q-SNN) that applies quantization to both synaptic weights and membrane potentials. By significantly compressing these two key elements, the proposed Q-SNNs substantially reduce both memory usage and computational complexity. Moreover, to prevent the performance degradation caused by this compression, we present a new Weight-Spike Dual Regulation (WS-DR) method inspired by information entropy theory. Experimental evaluations on various datasets, including static and neuromorphic, demonstrate that our Q-SNNs outperform existing methods in terms of both model size and accuracy. These state-of-the-art results in efficiency and efficacy suggest that the proposed method can significantly improve edge intelligent computing.

## CCS CONCEPTS

• **Computing methodologies** → **Computer vision tasks**.

## KEYWORDS

Spiking Neural Networks, Neuromorphic Datasets, Quantization

## 1 INTRODUCTION

Brain-inspired spiking neural networks (SNNs) have emerged as a promising approach for the next generation of machine intelligence. In contrast to traditional deep neural networks (DNNs), the spiking neuron in SNNs transmits information through sparse and binary spikes, which converts the computationally intensive multiply-accumulate (MAC) operations into computationally efficient accumulate (AC) operations. This energy-efficient characteristic of SNNs has triggered a growing interest in the design of neuromorphic hardware, including SpiNNaker [30], TrueNorth [1], Loihi [9], and Tianjic [31], etc. These neuromorphic hardware draw upon the storage and computing paradigms of the human brain, enabling the energy efficiency advantages of SNNs to be further demonstrated. However, despite the energy efficiency of SNNs, their performance in practical applications requires improvement.

In recent years, numerous studies have dedicated efforts to crafting expansive and complex network architectures for SNNs, leading to notable enhancements in performance across various challenging tasks such as image classification [16, 48], language generation [3, 53], and object recognition [20, 41]. Despite their commendable performance, these studies come at the cost of massive model parameters and high computational complexity. As a consequence, these large-scale SNNs sacrifice the inherent energy efficiency advantages that are closely associated with SNNs, thereby presenting challenges for their efficient deployment on real-world resource-constrained edge devices.

There is a growing body of research investigating compression techniques for large-scale SNNs. These techniques include pruning [7, 11, 49], knowledge distillation [23, 42, 47], neural architecture search [27, 33], and quantization [11, 18, 19, 50]. Quantization techniques have attracted considerable interest due to their ability to convert synaptic weights from a high-precision floating-point representation to a low bit-width integer representation. This conversion facilitates efficient deployment on resource-constrained devices by enabling the use of lower-precision arithmetic units, which are both cheaper and more energy-efficient. Binarization within this domain stands out for its hardware-friendly feature, which restricts data to two possible values (-1 and +1), leading to significant efficiency in terms of storage and computation. However, existing binarization approaches in SNNs focus solely on quantizing synaptic weights, overlooking the memory-intensive aspect associated with membrane potentials [18, 32, 34, 38, 43]. Therefore, there remains room for further efficiency optimizations. While prior lightweight research has explored the quantization of membrane potentials [50], the synaptic weights in this work remain non-binary, which results in a performance gap compared to full-precision SNN models.

This paper introduces a lightweight spiking neural network (SNN) architecture called Quantized SNN (Q-SNN) which prioritizes energy efficiency while maintaining high performance. We achieve this by quantizing two key elements within the network, namely synaptic weights and membrane potentials. This targeted quantization aims to maximize the energy efficiency of the proposed architecture. In addition, to further enhance performance within Q-SNNs, we leverage the principles of information entropy by introducing a novel Weight-Spike Dual Regulation (WS-DR) method. This method aims to maximize the information content within Q-SNNs, ultimately leading to improved accuracy. The main contributions of this work are summarized as follows:

- We introduce a novel SNN architecture, called Q-SNN, designed for efficient hardware implementation and low energy consumption. Q-SNN achieves this goal by employing the quantization technique on two key elements of the network: (1) synaptic weights using binary representation and (2) membrane potentials using low bit-width representation. This targeted quantization significantly improves the efficiency of the network.

- We analyze how to enhance the performance of Q-SNNs from the theory of information entropy and propose a novel Weight-Spike Dual Regulation (WS-DR) method. Inspired by information entropy theory, WS-DR adjusts the distribution of both weights and spikes within the network. This method maximizes the information content within Q-SNNs and results in improved accuracy.
- We conduct comprehensive experiments on various benchmark datasets, including static and neuromorphic, to validate the efficiency and effectiveness of our method. Experimental results demonstrate that our method achieves state-of-the-art results in terms of both efficiency and performance, underscoring its capability to boost the development of edge intelligent computing.

## 2 RELATED WORK

Several compression techniques have been explored to address the challenges associated with large-scale Spiking Neural Networks (SNNs). These techniques include pruning, knowledge distillation, and neural architecture search. Pruning involves removing redundant parameters by eliminating unnecessary nodes or branches within the network. Existing pruning approaches for SNNs can be broadly classified into two categories. The first category leverages established pruning methods developed for Deep Neural Networks (DNNs) and applies them to SNNs in both spatial and temporal domains [8, 11, 49]. The second category draws inspiration from biological processes in the human brain. These bio-inspired pruning algorithms model the synaptic regrowth process to achieve network compression [5, 6]. Knowledge distillation is a technique that enables the transfer of knowledge from a larger pre-trained model to a smaller model. Existing knowledge distillation methods for SNNs can also be grouped into two main categories. The first category uses the knowledge from pre-trained ANNs or SNNs to guide the training process of smaller student SNN models [23, 42, 47]. However, these methods require the additional computational overhead of training a large-scale teacher model. The second category, referred to as self-distillation [13, 14], eliminates the need for a separate teacher model. In self-distillation, the model acts as its own teacher and generates the necessary supervisory signals for training. This approach reduces computational costs by eliminating the requirement for training an additional teacher model.

Neural Architecture Search (NAS) is an automated approach for devising high-performing neural network structures under resource constraints. Early NAS applications in SNNs were limited by their focus on exploring pre-defined network blocks, thereby hindering the identification of potentially optimal designs beyond these predefined constraints [29]. To address this limitation, researchers have developed algorithms to search for optimal network structures across the entire network. These advancements aim to enable the design of more flexible and potentially more efficient SNN models [4, 21, 27].

Quantization is a powerful technique for compressing SNN models. Unlike traditional ANNs which use real-valued activations, SNNs communicate through binary spikes. Consequently, the main energy consumption in SNNs is attributed to the use of floating-point synaptic weights. To address this issue, existing quantized

SNNs mainly focus on reducing the bit-width of weights, typically down to 2, 4, or 8 bits [7, 11, 40]. However, further reduction to a single bit, i.e. binary weights, often leads to significant performance degradation or even failure of training convergence. This challenge has motivated research on specifically designed binary SNNs. For instance, Wang et al. [43] proposed a weights-thresholds balance conversion method to achieve SNNs with binary weights. Qiao et al. [34] used the surrogate gradient (SG) method to train binary SNNs for efficient processing of event-based data. Building upon this work, Jiang et al. [18] introduced a Bayesian-based learning algorithm that demonstrates superior accuracy and calibration compared to the SG method for binary SNNs. In addition, Kheradpisheh et al. [19] proposed a temporal-based binary SNN where each neuron fires at most once with learning taking place only upon spike emission. While these binary SNN approaches offer significant efficiency advantages, they still have limitations. Firstly, these methods focus solely on quantizing synaptic weights, neglecting the memory footprint associated with membrane potentials. This presents an opportunity for further efficiency improvements. Secondly, binary SNNs often suffer from severe information loss during inference, resulting in a substantial performance gap when compared to their full-precision SNN counterparts.

In order to overcome these limitations, we introduce the Quantized SNN (Q-SNN) that quantizes both synaptic weights and membrane potentials to maximize efficiency. Furthermore, to mitigate the performance gap between Q-SNN and full-precision SNNs, we leverage the principles of information entropy and propose a novel Weight-Spike Dual Regulation (WS-DR) method. By integrating WS-DR, we can train efficient and high-performance Q-SNNs from scratch, paving the way for deploying SNNs on resource-constrained devices.

## 3 PRELIMINARY

### 3.1 Leaky Integrate-and-Fire model

SNNs rely on spiking neurons as their fundamental processing units. These models aim to replicate the information processing capabilities of biological neurons. Several prominent examples include the Hodgkin-Huxley [15], Izhikevich [17], and Leaky Integrate-and-Fire (LIF) models [45]. Due to its computational efficiency, we adopt the LIF model. Its membrane potential, a key element of a neuron's firing behavior, is mathematically described as,

$$u_i^l[t] = \tau u_i^l[t-1] + \sum_j w_{ij}^l s_j^{l-1}[t], \tag{1}$$

where $\tau$ is the constant leaky factor, $w_{ij}^l$ is the synaptic weight between neuron $j$ in layer $l-1$ and neuron $i$ in layer $l$, and $s_j^{l-1}[t]$ is the input spike from presynaptic neuron $j$ at time $t$. Neuron $i$ integrates inputs and emits a spike when its membrane potential exceeds the firing threshold. Mathematically, the spike generation function is stated as,

$$s_i^l[t] = \begin{cases} 1, & \text{if } u_i^l[t] \geq \theta, \\ 0, & \text{otherwise,} \end{cases} \tag{2}$$

where $\theta$ denotes the firing threshold parameter. Following each spike emission, the spiking neuron $i$ undergoes a reset mechanism

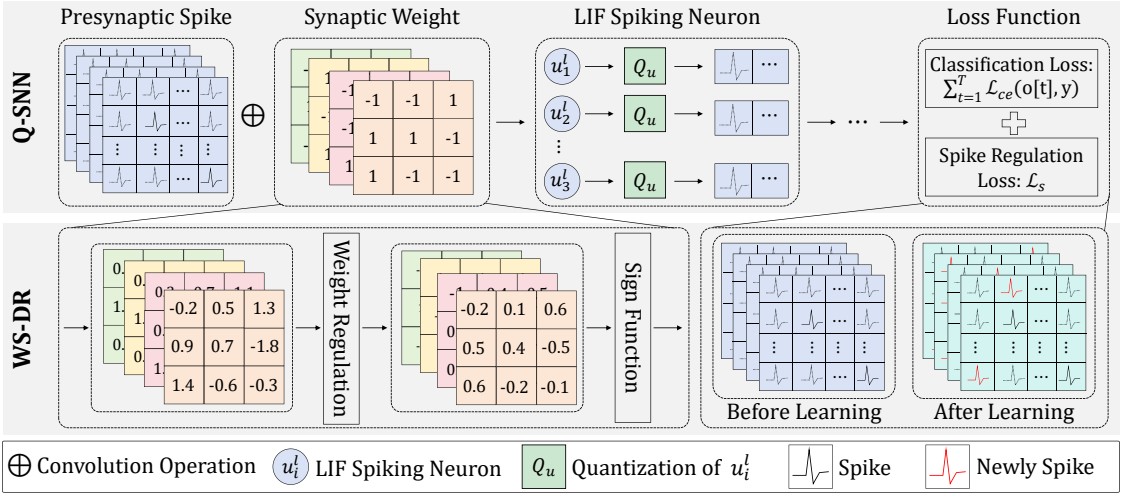

**Figure 1: The overall workflow of the proposed Q-SNN.**

that updates its membrane potential. This reset process is mathematically defined as,

$$u_i^l[t] = u_i^l[t] \cdot \left(1 - s_i^l[t]\right). \tag{3}$$

This work uses the hard reset mechanism where the membrane potential of neuron $i$ is reset to zero upon emitting a spike and remains unchanged in the absence of a spike.

## 3.2 Surrogate gradient learning method

Training of SNNs requires calculating the gradient of the loss function with respect to the synaptic weight. The chain rule provides a powerful tool to achieve this. By applying the chain rule, the derivative of the loss function $L$ with respect to the synaptic weight $w_{ij}^l$ can be decomposed into the following equation,

$$\frac{\partial L}{\partial w_{ij}^l} = \sum_{t=1}^T \left( \frac{\partial L}{\partial s_i^l[t]} \frac{\partial s_i^l[t]}{\partial u_i^l[t]} \frac{\partial u_i^l[t]}{\partial w_{ij}^l} + \frac{\partial L}{\partial u_i^l[t+1]} \frac{\partial u_i^l[t+1]}{\partial u_i^l[t]} \frac{\partial u_i^l[t]}{\partial w_{ij}^l} \right). \tag{4}$$

However, training SNNs presents a distinct challenge compared to traditional ANNs and Deep Neural Networks (DNNs) due to the non-differentiable nature of the spiking (i.e. firing) mechanism. Specifically, the term $\partial s_i^l[t]/\partial u_i^l[t]$ represents the gradient of the spike generation function (described in Eq. 2). This function evaluates to infinity at the moment of spike emission and to zero elsewhere, making it incompatible with the traditional error backpropagation used in ANN/DNN training. To tackle this issue, existing studies employ surrogate gradients to approximate the true gradient [45]. These surrogate gradients take various shapes, including rectangular [46], triangular [12], and linear [44]. In this paper, we use the triangular-shaped surrogate gradient which is mathematically defined as,

$$\frac{\partial s_i^l[t]}{\partial u_i^l[t]} = max\left(0, \beta - \left|u_i^l[t] - \theta\right|\right), \tag{5}$$

where $\beta$ is the factor that defines the range of gradient computation.

## 4 METHOD

This section introduces the core aspects of our work. First, we present the design of the proposed lightweight Q-SNN architecture, focusing on the key element of quantizing both synaptic weights and membrane potentials for improving efficiency. Then, we explore methods for enhancing its performance through an information theory-based analysis. Leveraging the principles of information entropy, we present a novel Weight-Spike Dual Regulation (WS-DR) method to maximize the information content within Q-SNNs, ultimately leading to improved accuracy.

## 4.1 Lightweight spiking neural networks

In order to exploit the energy efficiency benefit inherent to SNNs, we introduce a Quantized SNN (Q-SNN). As shown in Figure 1, the first characteristic of the proposed Q-SNNs is to quantize the synaptic weight. The synaptic weights in SNNs are commonly represented as 32-bit values, which often encounter challenges such as large storage demands, increased computational complexity, and high power consumption. Therefore, in Q-SNNs, we quantize the synaptic weight into a 1-bit representation, which is formulated as,

$$Q_w(w) = \alpha_w \cdot sign(w), \tag{6}$$

$$sign(w) = \begin{cases} +1, & \text{if } w \geq 0, \\ -1, & \text{otherwise,} \end{cases} \tag{7}$$

where $w$ indicates the 32-bit weight, $sign(w)$ represents the binarization operation of obtaining the 1-bit weight, and $\alpha_w$ denotes the scaling factor for synaptic weights. The binarization operation $sign(w)$ typically suffers from severe information loss, resulting in performance degradation of Q-SNNs. To overcome this issue, a channel-wise scaling factor $\alpha_w$ is introduced to mitigate the impact of information loss, and it is calculated as the average of the absolute value of weights in each output channel [35].

The second characteristic of the proposed Q-SNNs is to quantize the memory-intensive membrane potentials in SNNs. As synaptic weights are reduced to a specific bit width, the membrane potential

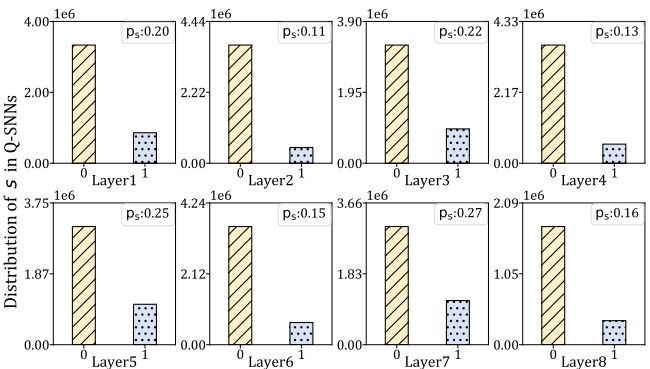

**Figure 2: The distribution of spikes $s$ in Q-SNNs, we select the first eight layers in ResNet-19 on CIFAR-10 for display.**

begins to emerge as the predominant factor in memory storage and computational overhead [50]. Therefore, the membrane potential plays a critical role in improving efficiency, but it is neglected in existing binary SNNs. Building upon this, we quantize the membrane potential within Q-SNNs to a low bit-width integer, with the quantization function described as,

$$Q_u(u) = \frac{\alpha_u}{2^k - 1} round\left(\left(2^k - 1\right) clip\left(\frac{u}{\alpha_u}, -1, 1\right)\right), \quad (8)$$

$$clip(x, -1, 1) = min\left(max\left(x, -1\right), 1\right), \quad (9)$$

where $u$ is the 32-bit membrane potential, $k$ denotes the number of bits assigned to the quantized integer, $\alpha_u$ is the scaling factor for membrane potentials, $round(\cdot)$ is a rounding operator, and $clip(\cdot)$ is a clipping operator saturating $x$ within the range $[-1, 1]$. In this paper, $k$ is set to 2/4/8, and a layer-wise scaling factor $\alpha_u$ is determined as the maximum value in each layer.

By integrating Eq. 6 and Eq. 8 into Eq. 1, the membrane potential of the LIF neuron model can be represented as,

$$u_i^l[t] = \tau \cdot Q_u\left(u_i^l[t-1]\right) + \sum_j \alpha_w \left(sign\left(w_{ij}^l\right) \oplus s_j^{l-1}[t]\right). \quad (10)$$

In this equation, it can be observed that the membrane potential $u_i^l[t-1]$ is quantized to a low bit-width, thereby requiring diminished storage demands. Furthermore, through the integration of binary spikes and binary weights, the proposed Q-SNNs can leverage cost-effective bitwise operations, i.e., $\oplus$, to execute the convolutional operation, thereby attaining enhanced computational efficiency in the inference process. In summary, the proposed Q-SNN architecture maximizes the efficiency benefit inherent to SNNs.

### 4.2 Analysis of information content in Q-SNNs

While the proposed Q-SNNs exhibit significant energy efficiency, it must be acknowledged that their task performance lags significantly behind that of full-precision SNNs. This performance gap can be attributed to the limited information representation capability of Q-SNNs due to the low-precision weights and spiking activities. To address this challenge, we explore into the concept of information entropy and analyze how it can be leveraged to enhance the performance of Q-SNNs.

In Q-SNNs, the synaptic weight and spike activity are binary values, i.e., $w \in \{-1, 1\}$ and $s \in \{0, 1\}$, both of them follow the Bernoulli distribution. Taking the binary spike $s$ as an example for analysis, its probability mass function can be expressed as,

$$f(s) = \begin{cases} p_s, & \text{if } s = 1, \\ 1 - p_s, & \text{if } s = 0, \end{cases} \quad (11)$$

where $p_s \in (0, 1)$ is the probability of $s$ being value 1. We measure the information content carried by $s$ using the theory of information entropy, expressed as,

$$\mathcal{H}(s) = -\left[p_s \ln(p_s) + (1 - p_s)\ln(1 - p_s)\right]. \quad (12)$$

From this equation, it can be observed that when $p_s$ approaches either 0 or 1, the entropy function $\mathcal{H}(s)$ may tend towards its minimum value 0. To evaluate the information content carried by $s$ in Q-SNNs, we conduct experiments with ResNet-19 on the CIFAR-10 dataset to obtain $p_s$ in each layer, as illustrated in Figure 2. Unfortunately, $p_s$ in each layer tends towards 0, resulting in severely limited information content carried by $s$. Even worse, the weight in Q-SNNs also faces the same challenge.

To enhance the performance of Q-SNN, increasing the information content of its binary weights and spike activities is necessary. Therefore, we analyze what condition $p_s$ satisfies so that the information entropy of binary values is maximized. Maximizing entropy in this context helps ensure that the quantized weight and membrane potential values are spread out as evenly as possible within their range. This uniform distribution helps to ensure that the quantized network's representational capacity is not overly constrained by the quantization process. It allows the quantized network to preserve as much information as possible despite the reduced precision of synaptic weights and membrane potentials in the network. To achieve this, we need to solve the following equation,

$$p^* = \arg\max_{p_s} \mathcal{H}(s). \quad (13)$$

By solving Eq. 13, we can determine $p^* = 0.5$. Therefore, we design a method to regulate the distributions of binary weights and spikes respectively, enabling their probability to be close to 0.5.

### 4.3 Weight-Spike dual regulation method

To mitigate the performance degradation caused by information loss in the quantization process, we propose a Weight-Spike Dual Regulation method (WS-DR) to increase the information content of weights and spike activities in Q-SNNs. As analyzed in Sec. 4.2, the variable obeying the Bernoulli distribution carries the maximum information content when $p = 0.5$. Therefore, in the following, we illustrate how to adjust the binary weight and spike activity to satisfy this condition as much as possible.

We first analyze the regulation for synaptic weights in Q-SNNs. The objective for the binary weight, denoted as $B_w$, to satisfy $p_w = 0.5$ implies that it has an equal probability of taking either 1 or -1. We experimentally observed that the float-point weight in layer $l$ before adopting binarization operation generally obeys a Gaussian distribution $\mathcal{N}(\mu_l, \sigma_l^2)$. However, the mean value $\mu_l$ and standard deviation $\sigma_l$ vary across layers. Inspired by this observation, we implement a normalization technique on float-point weights in

each layer, as detailed below,

$$\hat{W}^l = \frac{W^l - \mu_l}{\sigma_l}, \tag{14}$$

where $W^l$ denotes the tensor of float-point weights in layer $l$. This equation enables $\hat{W}^l$ to obey a standard normal distribution $\mathcal{N}(0, 1)$, with zero serving as the symmetry axis. By subsequently employing the *sign* function on $\hat{W}^l$, we can ensure $p_w = 0.5$.

We then analyze the regulation for spike activities in Q-SNNs. Similar to the binary weight $B_w$ in Q-SNNs, achieving the maximal information entropy of spike activities implies that each neuron in Q-SNNs emits a spike with a probability of 1/2 at all possible spike activity locations, i.e., the firing rate is 50%. This will seriously impair the sparsity characteristic of SNNs and sacrifice their inherent energy efficiency advantage. Therefore, in Q-SNNs, we design a loss function to impose a soft regulation on $s$. The goal of this loss function is to increase the information entropy of $s$ per layer as much as possible, rather than rigidly enforcing its probability to be 0.5. The proposed loss term $\mathcal{L}_s$ is defined as,

$$\mathcal{L}_s = \sum_{l=2}^{L-1} (f_l - 0.5)^2, \quad f_l = \frac{1}{N_l \times T} \left( \sum_{i=1}^{N_l} \sum_{t=1}^{T} s_i^l[t] \right), \tag{15}$$

where $L$ is the total number of layers in the network, $f_l$ denotes the firing rate of spiking neurons in layer $l$, $N_l$ is the number of neurons in layer $l$, and $T$ indicates the simulation time step. The proposed loss term $\mathcal{L}_s$ only regulates the spike distributions in hidden layers, disregarding the input and output layers as they are correlated with input data and classification outcomes. Consequently, the overall loss function is defined as,

$$\mathcal{L} = \frac{1}{T} \sum_{t=1}^{T} \mathcal{L}_{ce} (\mathbf{o}(t), \mathbf{y}) + \lambda \mathcal{L}_s, \tag{16}$$

where $\mathbf{o}[t]$ is the output of Q-SNNs at time $t$ and $y$ is the target label. $\mathcal{L}_{ce}$ is the cross-entropy loss for classification, $\mathcal{L}_s$ is the soft regulation term designed to increase the information content of $s$, and $\lambda$ is a hyperparameter that controls the contribution of $\mathcal{L}_s$. By integrating the normalization technique on $w$ and the soft regulation loss term on $s$, the weight and spike in Q-SNNs can carry more information content, thus mitigating the performance degradation caused by information loss during the quantization process. Finally, we summarize the training procedure of our method in Algorithm 1.

## 5 EXPERIMENTS

In this section, we first outline the experimental setup and implementation details of the proposed Q-SNN with the WS-DR method. We then conduct a comprehensive evaluation, comparing the performance and model size of our approach against existing quantized SNN techniques. Finally, we employ ablation studies to validate the efficiency and effectiveness of our method.

### 5.1 Experimental setup

*Datasets.* We evaluate our method on image classification tasks, encompassing both static image datasets like CIFAR-10 [22], CIFAR-100 [22], and TinyImageNet [10], alongside neuromorphic datasets

---

**Algorithm 1** Train a Q-SNN with the Weight-Spike Dual Regulation (WS-DR) method for one epoch.

**Input:** A Q-SNN model: $\mathcal{W} = \{W^1, \cdots, W^L\}$; The number of training iteration in one epoch: $I_{train}$; Training dataset.
**Output:** The Q-SNN model with updated weights.

1: **for** $i = 1 \rightarrow I_{train}$ **do**
2:     Get a minibatch of training data and target labels $(\mathbf{I}, \mathbf{Y})$;
3:     Initialize an empty list $F$;
4:     **for** $l = 1 \rightarrow L$ **do**
5:         Regulate the distribution of 32-bit weights: $\hat{W}^l = \frac{W^l - \mu_l}{\sigma_l}$;
6:         Compute 1-bit weights and channel-wise scaling factors:
        $B_w = sign(\hat{W}^l)$ , $\alpha_w = \frac{||\hat{W}^l||_1}{n}$;
7:         **for** $t = 1 \rightarrow T$ **do**
8:             Compute the membrane potential $(U^l[0] = 0, S^0[t] = \mathbf{I})$:
            $U^l[t] = \tau U^l[t-1] + \alpha_w \left( B_w \oplus S^{l-1}[t] \right)$;
9:             Compute a layer-wise scaling factor: $\alpha_u = max \left( U^l[t] \right)$;
10:           Quantize the membrane potential to $k$ bits:
          $U^l[t] \leftarrow \frac{\alpha_u}{2^k - 1} round \left( \left( 2^k - 1 \right) clip \left( \frac{U^l[t]}{\alpha_u}, -1, 1 \right) \right)$;
11:           Generate binary spikes via Heaviside step function:
          $S^l[t] = Heaviside \left( U^l[t] - \theta \right)$;
12:         **end for**
13:         $S^l = concat \left( S^l[1], \cdots, S^l[T] \right)$;
14:         Compute the firing rate: $f_l = \frac{1}{N_l \times T} \left( \sum_{i=1}^{N_l} \sum_{t=1}^{T} S_i^l[t] \right)$;
15:         $F \leftarrow F.append(f_l)$;
16:     **end for**
17:     Regulate the distribution of spikes: $\mathcal{L}_s = \sum_{l=2}^{L-1} (F[l] - 0.5)^2$;
18:     Compute the loss: $\mathcal{L} = \frac{1}{T} \sum_{t=1}^{T} \mathcal{L}_{ce} + \lambda \mathcal{L}_s$;
19:     Backpropagation and update model parameters;
20: **end for**

---

such as DVS-Gesture [2] and DVS-CIFAR10 [24]. These datasets hold substantial importance within the realms of machine learning and neuromorphic computing, serving as standard benchmarks for evaluating diverse methodologies. Before introducing the experiments, we briefly outline each dataset. The CIFAR-10 and CIFAR-100 are color image datasets, with each dataset containing 50,000 training images and 10,000 testing images. Each image features 3 color channels and a spatial resolution of 32×32 pixels. CIFAR-10 is composed of 10 categories, whereas CIFAR-100 comprises 100 categories. The TinyImageNet dataset is a subset of the ImageNet dataset, consisting of 200 categories, with each category containing 500 training images and 50 test images. Each image has 3 color channels and a spatial resolution of 64×64 pixels. The DVS-Gesture and DVS-CIFAR10 are neuromorphic datasets captured using Dynamic Vision Sensor (DVS) event cameras, both featuring a spatial resolution of 128×128. The DVS-Gesture dataset consists of 1464 samples, with 1176 allocated for training and 288 for testing. The DVS-CIFAR10 is the most challenging neuromorphic dataset, featuring 9,000 training samples and 1,000 testing samples. During the preprocessing process of the DVS-CIFAR10 dataset, we apply the data augmentation technique proposed in [26].

**Table 1: Classification performance comparison on both static image datasets and neuromorphic datasets.**

| Dataset | Method | Architecture | Learning | Bit Width | Timestep | Accuracy |
|---|---|---|---|---|---|---|
| CIFAR-10 | Full-Precision SNN[‡] | ResNet19 | Direct train | 32w-32u[1] | 2 | 96.36% |
| | Roy et al. [36] | VGG9 | ANN2SNN | 1w-32u | - | 88.27% |
| | Rueckauer et al. [37] | 6Conv3FC | ANN2SNN | 1w-32u | - | 88.25% |
| | Wang et al. [43] | 6Conv3FC | ANN2SNN | 1w-32u | 100 | 90.19% |
| | Yoo et al. [51] | VGG16 | ANN2SNN | 1w-32u | 32 | 91.51% |
| | Deng et al. [11] | 7Conv3FC | Direct train | 1w-32u | 8 | 89.01% |
| | Pei et al. [32] | 5Conv1FC | Direct train | 1w-32u | 1 | 92.12% |
| | Zhou et al. [52] | VGG16 | Direct train | 2w-32u | - | 90.93% |
| | Yin et al. [50] | ResNet19 | Direct train | 2w-2u | 4 | 90.79% |
| | **Proposed Q-SNN** | ResNet19 | Direct train | 1w-8u | 2 | **95.54%** |
| | | | | 1w-4u | 2 | **95.31%** |
| | | | | 1w-2u | 2 | **95.20%** |
| CIFAR-100 | Full-Precision SNN[‡] | ResNet19 | Direct train | 32w-32u | 2 | 79.52% |
| | Roy et al. [36] | VGG16 | ANN2SNN | 1w-32u | - | 54.44% |
| | Lu et al. [28] | VGG15 | ANN2SNN | 1w-32u | 400 | 62.07% |
| | Wang et al. [43] | 6Conv2FC | ANN2SNN | 1w-32u | 300 | 62.02% |
| | Yoo et al. [51] | VGG16 | ANN2SNN | 1w-32u | 32 | 66.53% |
| | Deng et al. [11] | 7Conv3FC | Direct train | 1w-32u | 8 | 55.95% |
| | Pei et al. [32] | 6Conv1FC | Direct train | 1w-32u | 1 | 69.55% |
| | **Proposed Q-SNN** | ResNet19 | Direct train | 1w-8u | 2 | **78.77%** |
| | | | | 1w-4u | 2 | **78.82%** |
| | | | | 1w-2u | 2 | **78.70%** |
| TinyImageNet | Full-Precision SNN[‡] | VGG16 | Direct train | 32w-32u | 4 | 56.77% |
| | Yin et al. [50] | VGG16 | Direct train | 8w-8u | 4 | 50.18% |
| | | | Direct train | 4w-4u | 4 | 49.36% |
| | | | Direct train | 2w-2u | 4 | 48.60% |
| | **Proposed Q-SNN** | VGG16 | Direct train | 1w-8u | 4 | **55.70%** |
| | | | | 1w-4u | 4 | **55.20%** |
| | | | | 1w-2u | 4 | **55.04%** |
| DVS-Gesture | Full-Precision SNN[‡] | VGGSNN[*] | Direct train | 32w-32u | 16 | 97.83% |
| | Pei et al. [32] | 5Conv1FC | Direct train | 1w-32u | 20 | 94.63% |
| | Qiao et al. [34] | 2Conv2FC | Direct train | 1w-32u | 150 | 97.57% |
| | Yoo et al. [51] | 15Conv1FC | Direct train | 1w-32u | 16 | 97.57% |
| | **Proposed Q-SNN** | VGGSNN | Direct train | 1w-8u | 16 | **97.92%** |
| | | | | 1w-4u | 16 | **97.57%** |
| | | | | 1w-2u | 16 | **96.53%** |
| DVS-CIFAR10 | Full-Precision SNN[‡] | VGGSNN | Direct train | 32w-32u | 10 | 82.10% |
| | Qiao et al. [34] | 2Conv2FC | Direct train | 1w-32u | 25 | 62.10% |
| | Pei et al. [32] | 5Conv1FC | Direct train | 1w-32u | 10 | 68.98% |
| | Yoo et al. [51] | 16Conv1FC | Direct train | 1w-32u | 16 | 74.70% |
| | **Proposed Q-SNN** | VGGSNN | Direct train | 1w-8u | 10 | **81.60%** |
| | | | | 1w-4u | 10 | **81.50%** |
| | | | | 1w-2u | 10 | **80.00%** |

[1]32w-32u: The model with 32-bit weights and 32-bit membrane potentials.

[‡]: Self-implementation results with the same experimental setting. [*]VGGSNN: 8Conv1FC.

*Implementation details.* We first present the architecture employed for each dataset. For static CIFAR-10 and CIFAR-100 datasets, we employ the well-established structure of ResNet19. For the TinyImageNet dataset, we employ the VGG16 structure to facilitate comparison with [50]. For DVS-Gesture and DVS-CIFAR10 datasets, we implement the structure of VGGSNN commonly used in SNNs [12, 39]. Within these architectures, the weights in the first layer and the last layer are quantized to 8 bit, while the weights in hidden layers are quantized to 1 bit [25]. Subsequently, we elucidate the implementation details within our experiments. We employ the Adam optimizer for the TinyImageNet dataset while the SGD optimizer for other datasets, and the batch size is set to 256 for

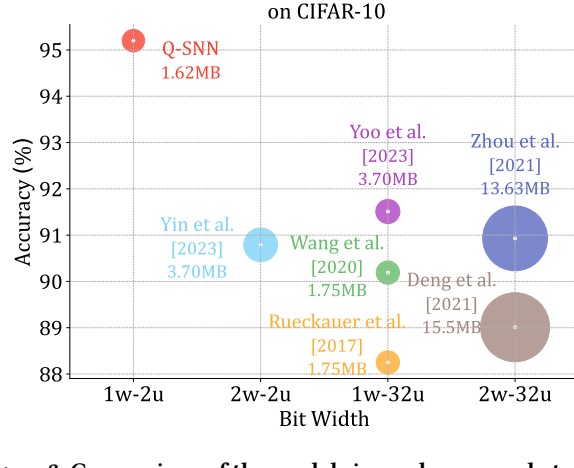

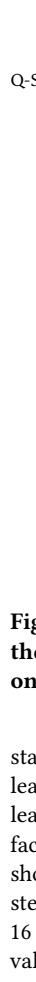

Figure 3: Comparison of the model size and accuracy between the proposed Q-SNN and existing quantized SNN approaches on the CIFAR-10 dataset.

static image datasets and 64 for neuromorphic datasets. The initial learning rate is set to 0.1 across all datasets and we adopt a cosine learning rate decay schedule during training. Moreover, the leaky factor $\tau$ and the firing threshold $\theta$ of the LIF neuron within Q-SNNs should be specified, which is set to 0.5 and 1, respectively. The time step $T$ is set to 2 for CIFAR-10 and CIFAR-100, 4 for TinyImageNet, 16 for DVS-Gesture, and 10 for DVS-CIFAR10. The hyperparameter value for the loss term $\mathcal{L}_s$ is set to 1e-3 for all utilized datasets. All experiments conducted in this paper leverage the PyTorch library, which is a versatile framework widely adopted in the field of deep learning research.

## 5.2 Comparative study

We compare the performance and model size of Q-SNNs with existing quantized SNN methods to prove the effectiveness and efficiency of our method, respectively. Our experiments across all datasets involve three bit-width configurations: 1w-8u, 1w-4u, and 1w-2u, where '1w-8u' signifies that the Q-SNN utilizes 1-bit synaptic weights and 8-bit membrane potentials.

We first analyze the performance comparison results. As shown in Table 1, when compared with ANN2SNN learning algorithms that necessitate a large time step to guarantee lossless conversion (i.e., Wang et al. [43] set it to 100 on CIFAR-10 and 300 on CIFAR-100), the proposed Q-SNNs trained from scratch require fewer time steps to attain top-1 performance, such as 2 on CIFAR datasets and 4 on TinyImageNet dataset. When compared with direct learning algorithms, our work also achieves the top-1 accuracy on all datasets, i.e., 95.54% on CIFAR-10 under 1w-8u, 78.82% on CIFAR-100 under 1w-4u, 55.70% on TinyImageNet under 1w-8u, 97.92% on DVS-Gesture under 1w-8u, and 81.60% on DVS-CIFAR10 under 1w-8u. Moreover, it can be observed from Table 1 that the proposed Q-SNNs demonstrate comparable performance to that of full-precision SNNs with extremely compressed bit width across all datasets. Noteworthy, on the DVS-Gesture dataset, Q-SNN configured with 1w-8u achieves an accuracy of 97.92%, surpassing the corresponding full-precision SNN model. This superiority can be

Table 2: Comparison of the memory footprint between the proposed Q-SNNs and full-precision SNNs. The value in brackets is the reduction in memory footprint relative to full-precision SNNs.

| Batch | Bit Width | ResNet19 | VGG16 | VGGSNN |
|---|---|---|---|---|
| batch=1 | 32w-32u | 50.80 (-0.00%) | 59.16 (-0.00%) | 37.44 (-0.00%) |
| | 1w-8u | 1.75 (-96.56%) | 1.96 (-96.69%) | 1.31 (-96.50%) |
| | 1w-4u | 1.68 (-96.69%) | 1.93 (-96.74%) | 1.25 (-96.66%) |
| | 1w-2u | 1.65 (-96.75%) | 1.91 (-96.77%) | 1.21 (-96.77%) |
| batch=64 | 32w-32u | 83.83 (-0.00%) | 75.67 (-0.00%) | 70.47 (-0.00%) |
| | 1w-8u | 10.0 (-88.07%) | 6.09 (-91.95%) | 9.57 (-86.42%) |
| | 1w-4u | 5.81 (-93.07%) | 3.99 (-94.73%) | 5.38 (-92.37%) |
| | 1w-2u | 3.72 (-95.56%) | 2.94 (-96.11%) | 3.28 (-95.35%) |
| batch=256 | 32w-32u | 184.49 (-0.00%) | 126.01 (-0.00%) | 171.13 (-0.00%) |
| | 1w-8u | 35.17 (-80.94%) | 18.67 (-85.18%) | 34.74 (-79.70%) |
| | 1w-4u | 18.40 (-90.03%) | 10.28 (-91.84%) | 17.96 (-89.51%) |
| | 1w-2u | 10.01 (-94.57%) | 6.09 (-95.17%) | 9.57 (-94.41%) |

attributed to the dataset's simplicity and abundant noise, where the Q-SNN utilizing a low bit-width representation demonstrates robustness.

In addition to the outstanding performance, our method also demonstrates a compact model size. We evaluate the model size of the proposed Q-SNN alongside existing quantized SNN methods on the CIFAR-10 dataset, with the results illustrated in Figure 3. Clearly, Q-SNN stands out with the smallest model size of 1.62MB and the highest accuracy of 95.20%. When compared with the work [50] that employs the same architecture of ResNet19, our method demonstrates a remarkable 56.22% reduction in model size and a 4.41% increase in accuracy with fewer time steps. In conclusion, our work further exploits the efficiency advantage inherent to SNNs while upholding superior performance, offering substantial advantages and potential for flexible deployment in real-world resource-limited scenarios.

## 5.3 Ablation study

We conduct ablation experiments to validate the efficiency of Q-SNN and the effectiveness of the WS-DR method, respectively. All ablation experiments are performed on the CIFAR-10 dataset with the architecture of ResNet19, and experimental setups follow the description in Sec. 5.1.

We first validate the efficiency of the proposed Q-SNN. As illustrated in Table 2, we compute the memory footprint of our method under different architectures and bit-width configurations, obtaining two conclusions. Firstly, the proposed Q-SNN typically realizes a significant reduction in memory footprint compared to full-precision SNNs. For instance, employing the ResNet19 architecture with the batch size of 1, the Q-SNN with 1w-8u, 1w-4u, and 1w-2u achieves memory footprint reduction of 96.56%, 96.69%, and 96.75% respectively when compared with the full-precision SNN. Secondly, the benefit of the membrane potential quantization is positively correlated with the batch size. Taking VGGSNN as an example, when the batch size is set to 1, the compression of the membrane potential from 8 bit to 2 bit only yields a memory footprint reduction from 96.50% to 96.77%. In contrast, when the batch size is

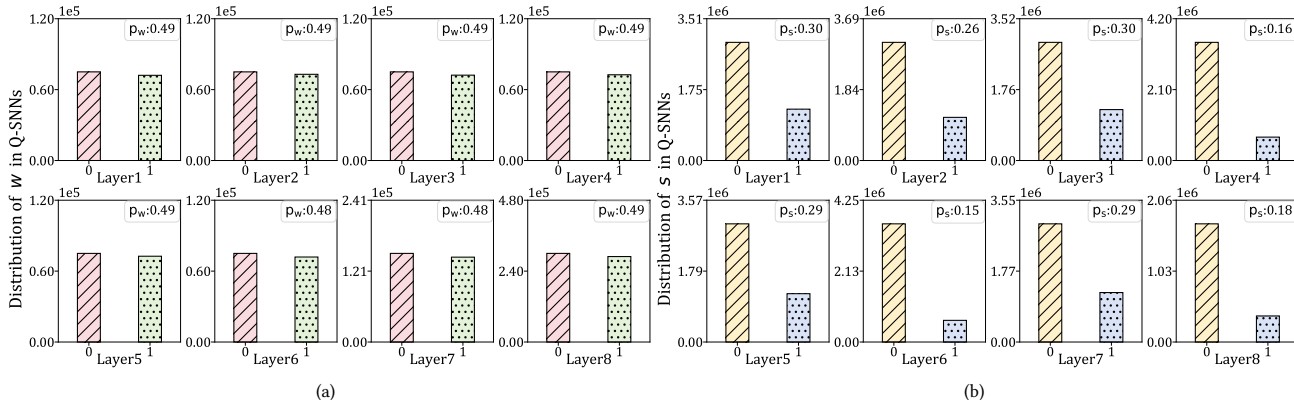

(a)                                                                      (b)

**Figure 4: (a) The distribution of synaptic weights in Q-SNN after applying the WS-DR method. (b) The distribution of spike activities in Q-SNN after applying the WS-DR method. These subfigures are plotted based on obtained results in the first eight layers of ResNet19 on the CIFAR-10 dataset.**

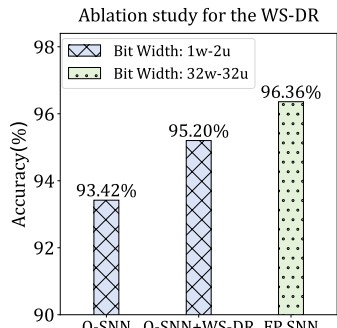

**Figure 5: Ablation study for the WS-DR method, where 'FP SNN' denotes the full-precision SNN.**

set to 256, the corresponding memory footprint reduction increases from 79.70% to 94.41%, underscoring the importance of membrane potential quantization. In summary, the proposed Q-SNNs have maximized the energy efficiency of the network by considering the quantization of both synaptic weights and memory-intensive membrane potentials.

We now validate the effectiveness of the proposed WS-DR method. As depicted in Figure 4, we plot the distribution of weights and spike activities in the first eight layers of ResNet-19 after applying the WS-DR method, respectively. It can be observed from in Figure 4(a) that the synaptic weights display a uniform distribution, with the probability $p_w$ approaching 0.5, so the $w$ in Q-SNNs have carried a greater amount of information. Moreover, it can be seen from Figure 4(b) that the probability $p_s$ becomes larger than Figure 2, thus also demonstrating the increased information content for $s$ in Q-SNNs. In addition to the enhanced information content, we also compare the performance of three models: pure Q-SNN, Q-SNN integrated with the WS-DR method, and full-precision SNN. As illustrated in Figure 5, under the configuration of 1w-2u, the Q-SNN integrated with the WS-DR method achieves an accuracy of 95.20%, surpasses that of Q-SNN without using the WS-DR method by 1.78%.

Notably, the performance of our model is comparable to the full-precision SNN of 96.36% by only using 1-bit synaptic weights and 2-bit membrane potentials. Therefore, the WS-DR method enhances the information content of both synaptic weights $w$ and spike activities $s$ in Q-SNNs, thereby resulting in performance comparable to that of full-precision SNN.

## 6 CONCLUSION

Spiking neural networks have emerged as a promising approach for next-generation artificial intelligence due to their sparse, event-driven nature and inherent energy efficiency. However, the prevailing focus within the SNN research community on achieving high performance by designing large-scale SNNs often overshadowed the energy-efficiency benefits inherent to SNNs. While efforts have been made to compress such large-scale SNNs by quantizing synaptic weights to lower bit widths, existing methods faced two main limitations. Firstly, they tend to overlook the memory footprint associated with membrane potentials in SNNs, leaving scope for further efficiency improvements. Secondly, the employed quantization on weights often leads to performance degradation of the quantized SNNs compared to their full-precision counterparts. This work introduced an efficient and effective Quantized SNN (Q-SNN) architecture to address these limitations. The proposed Q-SNN used efficient 1-bit weights and low-precision (2/4/8-bit) membrane potentials to further maximize SNN efficiency. In addition, inspired by information entropy theory, we proposed a novel Weight-Spike Dual Regulation (WS-DR) method to mitigate the performance degradation caused by quantization. This enabled the training of high-performance Q-SNNs from scratch. Experimental evaluations on static and neuromorphic datasets demonstrated that the proposed Q-SNN with WS-DR achieved state-of-the-art results in efficiency and performance compared to existing quantized SNN approaches. The obtained results demonstrate the potential of the proposed Q-SNNs in facilitating the widespread deployment of neuromorphic intelligent systems and advancing the development of edge intelligent computing.

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
