# OpenReview forum: "Q-SNNs: Quantized Spiking Neural Networks"
_acmmm.org/ACMMM/2024/Conference — MM2024 Poster_

### Official Review · Reviewer_SMM4 · 2024-05-24

**Rating:** 5
**Confidence:** 4

**Summary:**

This paper presents a novel approach to enhancing the efficiency of SNNs for machine intelligence applications. The authors introduce a lightweight SNN architecture that applies quantization to synaptic weights and membrane potentials, significantly reducing memory usage and computational complexity. To enhance accuracy attributed to the compression, the WS-DR regulation method is proposed, drawing on information entropy theory. The paper reports experimental evaluations on various datasets, demonstrating their superiority in model size and accuracy compared to existing methods.

**Strengths:**

1.This paper presents a quantization strategy applied to both weights and membrane potentials, significantly reducing the computational and memory footprints of SNNs. This advancement is particularly impactful given the increasing demand for efficient machine learning models.
2.Through extensive experimental evaluation, the application of information entropy theory to develop the WS-DR method has been proven effective. This method establishes a theoretical foundation for optimizing information-limited SNNs.
3.The authors conducted solid experiments, including evaluations of accuracy, model size, memory footprint, and various datasets, to demonstrate the effectiveness and energy efficiency of Q-SNNs.

**Limitations:**

1.Q-SNN performs quantization on weights and membrane potentials. So why does the WS-DR method aim to maximize the information entropy of weights and spikes rather than that of weights and membrane potentials?
2.When designing the soft regulation loss $L_s$, why did the authors choose to calculate it over the entire time window rather than based on each time step? Calculating the loss functon $L_s$ based on each time step to maximize the information entropy of each time, which seems to be more in line with the authors' motivation for designing WS-DR.

**Suitability:**

3

---

### Official Review · Reviewer_8qRv · 2024-05-24

**Rating:** 3
**Confidence:** 3

**Summary:**

The paper proposes the Q-SNN framework, focusing on synaptic weights and membrane potentials and introducing the Weighted Sum of Diverse Ranks (WSDR) based on information entropy, thereby advancing the field of quantized Spiking Neural Networks (SNNs).

**Strengths:**

The proposed models achieve decent accuracy with fewer time steps and smaller bit widths.

**Limitations:**

Limitations:
1.The selected architectures are relatively simple, while recent years have seen numerous transformer variants in the SNN field. The paper only mentions ResNet and VGG architectures.
2. Incomplete comparative experiments: There is a lack of comparison between Q-SNN and other quantization methods in the SNN field in terms of fixed bit width, datasets, architectures, etc.
3.The paper does not discuss dequantization methods, which are essential for quantization techniques.
4. There is no experimental or explanatory information about the parameter lambda and its role.
5. In terms of perspective, the author points out that other SNN models are primarily comparing accuracy, but it's worth noting that the author is also essentially comparing accuracy. It is suggested that the author include analyses of energy consumption, memory usage, parameter count, etc., rather than  focusing more on accuracy comparisons. The advantage of SNN lies in its energy efficiency, which the author has not adequately addressed in the paper.
6. The paper is a modification of the SNN mechanism and does not seem to include any content related to multimodal aspects.
Questions:
1.Regarding the bit width setting, if weights can be up to 1w, why can't membrane potential reach 1u?
2.How was the parameter count of 1.62M calculated? Please introduce in detail.

**Suitability:**

1

---

### Official Review · Reviewer_AgBZ · 2024-05-24

**Rating:** 5
**Confidence:** 4

**Summary:**

This paper proposes an efficient and effective Quantized Spiking Neural Network (Q-SNN) that compresses both weights and membrane potentials. Moreover, the authors introduce a Weight-Spike Dual Regulation method to mitigate performance degradation of Q-SNN. Additionally, experiments on various datasets demonstrate that the proposed approach achieves state-of-the-art results compared to existing quantized SNN approaches.

**Strengths:**

Clear Structure and Presentation: The paper is well-organized and presented in a manner that is easy to follow, enhancing the reader's understanding of the proposed method.

Information Entropy Theory: This paper utilizes information entropy theory to analyze the reasons for performance degradation in model quantization and proposes a novel Weight-Spike Dual Regulation method to effectively enhance information retain.

Performance and Model Size: The proposed Quantized SNN not only surpasses previous works in performance but also demonstrates a significantly more compact model size. This dual achievement in both efficiency and efficacy marks a notable advancement in the field.

**Limitations:**

Symbol Usage Issue: The same symbol \(u_i^l[t]\) is used on both sides of Eq.(3), which may lead to an inaccurate expression. It is recommended that the authors revise the notation to avoid confusion and improve the clarity of the expression.
Equation Expression Issue: The current expression of Eq.(8) may cause the quantized membrane potential to be \( k+1 \) bits instead of the intended \( k \) bits. This issue also appears in Algorithm 1. The authors should review and correct these formulas to ensure the correctness and consistency of the computations.
Impact of Soft Regulation Loss Function: For Q-SNN, this paper introduces a soft regulation loss function on \( s \) to increase the information entropy of the binary spike activities. However, does this regulation lead to a higher spike firing rate, thereby consuming more computational resources? It is suggested that the authors further investigate this issue and provide corresponding experimental results to prove it.

**Suitability:**

3

---

### Official Review · Reviewer_DuJw · 2024-06-07

**Rating:** 2
**Confidence:** 3

**Summary:**

This paper introduces a method to enhancing the efficiency of spiking neural networks for deployment in resource-constrained environments. The authors propose a quantized snn (Q-SNN) that utilizes quantization techniques on both synaptic weights and membrane potentials, significantly reducing memory usage and computational complexity. Additionally, the paper introduces the weight-spike dual regulation method, inspired by information entropy theory, to mitigate performance degradation due to quantization. The Q-SNN is evaluated on several datasets, demonstrating the effectiveness of the method.

**Strengths:**

1.The paper is well-organized, guiding the reader through the introduction, methodology.

2.The authors validate their method across several datasets, demonstrating the effectiveness.

**Limitations:**

1.While the paper demonstrates significant improvements, the practical implementation of the proposed Q-SNN and WS-DR is complex, potentially limiting its adoption in real-world applications.

2.Although the paper provides comparisons with existing methods, a more detailed analysis of how Q-SNNs fare against other compression techniques like pruning and knowledge distillation in various scenarios would strengthen the claims.

3.The scalability of Q-SNNs to very large-scale SNN models or more complex tasks beyond those tested is not thoroughly explored.

4.While WS-DR is introduced and shown to be effective, a more in-depth theoretical analysis and understanding of its workings and potential limitations would provide a clearer picture of its benefits and drawbacks.

5.The WS-DR introduces additional computational overhead during training. The paper does not quantify this overhead or compare it with the benefits gained, leaving a gap in understanding the trade-offs involved.

6.The paper does not explore the sensitivity of the results to different levels of quantization. Understanding how varying the bit-widths for weights and membrane potentials affects performance and efficiency would be important for practical implementation.

**Suitability:**

1

---

### Meta-Review · Area_Chair_WEso · 2024-06-30

**Recommendation:** Accept (Poster)
**Confidence:** 5

**Metareview:**

The paper proposes a novel Quantized Spiking Neural Network (Q-SNN) that quantizes synaptic weights and membrane potentials. According to the reviewers' feedback, acceptance of the paper is being considered. This method significantly reduces memory usage and computational complexity, making it suitable for resource-constrained and low-power edge devices. Furthermore, the paper introduces a novel Weight-Spike Dual Regulation (WSDR) method inspired by information entropy theory to ensure high performance of the model. Experimental results demonstrate its advantages in both model size and accuracy, highlighting its significant potential for edge intelligent computing.